# The Equine Dental Pulp: Histomorphometric Analysis of the Equine Dental Pulp in Incisors and Cheek Teeth

**DOI:** 10.3390/vetsci9060261

**Published:** 2022-05-30

**Authors:** Jessica Roßgardt, Laura Beate Heilen, Kathrin Büttner, Jutta Dern-Wieloch, Jörg Vogelsberg, Carsten Staszyk

**Affiliations:** 1Institute of Veterinary-Anatomy, -Histology and -Embryology, Faculty of Veterinary Medicine, Justus-Liebig-University Giessen, Frankfurter Strasse 98, 35390 Giessen, Germany; jessica.rossgardt@vetmed.uni-giessen.de (J.R.); laura.b.heilen@vetmed.uni-giessen.de (L.B.H.); jutta.dern-wieloch@vetmed.uni-giessen.de (J.D.-W.); joerg.vogelsberg@vetmed.uni-giessen.de (J.V.); 2Unit for Biomathematics and Data Processing, Faculty of Veterinary Medicine, Justus-Liebig-University Giessen, Frankfurter Strasse 95, 35390 Giessen, Germany; kathrin.buettner@vetmed.uni-giessen.de

**Keywords:** equine dental pulp, pulp cavity system, odontoblastic layer, predentin

## Abstract

To maintain a healthy and functional status, equine hypsodont teeth have to produce lifelong large amounts of subocclusal dentin to prevent occlusal pulp exposure, which is caused by occlusal wear. To examine the cyto- and histological components that guarantee the lifelong high productivity of equine pulp, a limited number of ten incisors and ten cheek teeth from seven adult horses (aged 5 to 24 years) and five foals were sampled for preliminary histomorphometric and histomorphological evaluations. Independently of age, the equine dental pulp featured constant layers of predentin and odontoblastic cells, as well as soft connective tissue, composed of a cellular fibrous matrix, in which blood vessels and nerve fibers were embedded. As a result of the progressive deposition of newly formed dentin, the layer of dentin became thicker with age, and the size of the pulp chamber decreased. In contrast to the brachydont teeth, the morphological characteristics of the odontoblastic layer and the width of the predentin layer did not change with age. Therefore, it is assumed that the equine pulp tissue retained their juvenile status, which explains its unchanged ability to produce high amounts of subocclusal dentin. These preliminary, but clinically significant, findings are worthy of further investigation in order to identify strategies for equine-specific endodontic therapies.

## 1. Introduction

Equine dental pulp (Pulpa dentis) shows an extreme functional adaptation to the specific biomechanical requirements acting on the hypsodont (high-crowned) teeth of horses. As a reaction to the significant dental wear caused by attrition (tooth–tooth contact) and abrasion (tooth–food contact) [1,2], equine teeth have to permanently seal their occlusal aspect of dental pulp using a distinct type of dentin, referred to as irregular secondary dentin [3,4], which appears as dark brown spots at the occlusal surface [4,5]. These so-called pulp positions are used as clinically relevant landmarks. Equine incisors contain up to two, and equine cheek teeth up to seven, pulp positions, which appear in a highly constant arrangement [6,7,8]. Underneath their occlusal surfaces, equine teeth feature a complex pulp cavity system. So-called pulp horns, i.e., finger-like protrusions of the pulp cavity, are present at varying distances, from less than one millimeter to several centimeters, beneath the pulp positions of the occlusal surface [5,9,10] (Figure 1).

In general, dental pulp is soft connective tissue composed of a fibrous matrix, in which various cells, blood vessels, and nerve fibers are embedded [11,12,13,14]. Mineralized dentin and its precursor, predentin, comprise a layer of odontoblastic cells. These prismatic odontoblasts are located in a narrow zone in the periphery of the dental pulp, where they line up in a palisade pattern [9,15]. Odontoblasts are the functional cells of the pulp system, as they produce dentin throughout their lives [9,13,14,16]. Their long cellular processes (also called Tomes fibers) pass through the dentinal tubules (Tubuli dentinales) up to the dentinoenamel junction or the dentinocemental junction [3,13]. Therefore, dental pulp and dentin are perceived as an interwoven morphological and functional unit, referred to as the dentin–pulp complex, or the endodontium [14,16,17,18]. The continual and significant dental wear on the occlusal surfaces of equine teeth, of several millimeters per year, even in older horses, requires the lifelong, compensating production of dentin in order to prevent the occlusal exposure of the dental pulp [4,19,20,21]. Besides the occlusal aspect of the pulp cavity, on the inner walls of the entire pulp cavity system, the progressive deposition of (pre-)dentin occurs. As a result, the dentin layer becomes increasingly thick with age, and the size of the pulp cavity decreases [9,22,23,24].

Preliminary examinations have focused on the spatial arrangement of equine dental pulp and its cavity system [25,26,27,28], but these investigations have neglected the histomorphometric changes to pulpal tissue with age. Therefore, the aim of this study was to describe and analyze equine pulpal tissue with regard to age-related changes and make recommendations for planning endodontic therapies.

## 2. Materials and Methods

Specimens were obtained from seven adult horses aged 5 to 24 years and five foals aged 2 days pre-parturition to 210 days post-parturition. All horses and foals died or were euthanized on grounds unrelated to this study at the Clinic of Maternity, Gynaecology, and Andrology and at the Clinic for Horse Surgery, Faculty of Veterinary Medicine, Justus Liebig University Giessen, Germany (Table 1).

### 2.1. Sampling

A postmortem macroscopic examination of the incisor and cheek teeth rows was carried out with regard to clinical health. Only clinically healthy deciduous and permanent incisors and molars were used for further histological examinations. Teeth showing signs of abnormal morphology and/or pathological changes (e.g., fractures, open pulp positions, infundibular decay) were excluded.

Before sampling, the horses were divided into three different age groups (AG):AG 1: 0 to 210 days.AG 2: 5 to 14 years.AG 3: 19 to 24 years.

Within 24 h of death, the horses’ heads were disarticulated, and the maxillary and mandibular cheek teeth rows were dissected using a water-cooled band saw (K440H, Kolbe Foodtec, Elchingen, Germany). Subsequently, all teeth were cleared of debris and stored in 10% buffered formalin (pH 7) at 4 °C.

### 2.2. Teeth

For this study, 10 incisors from the lower jaw (Triadan 301, 401) and 10 cheek teeth from the upper jaw (Triadan 108, 208) [29] were sectioned using a diamond-coated, water-cooled micro-band saw (MBS 240/E, Proxxon S.A., Wecker, Luxembourg). Horizontal sections were taken from three defined levels (Figure 2): Subocclusal (*so*).Central (*c*).Apical (*a*).

**Figure 2 vetsci-09-00261-f002:**
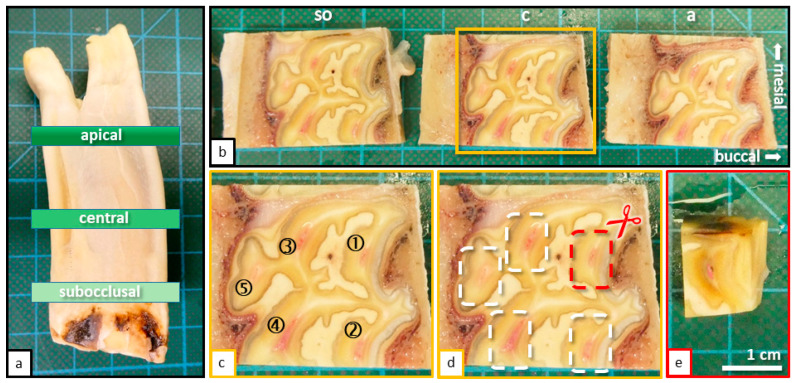
(**a**) Cheek tooth of an adult horse (208), sectioned horizontal levels (subocclusal, central, and apical) are indicated. (**b**) Sections of the tooth (208) displayed in Figure 1a, occlusal view, sectioned with a diamond coated micro-band saw. Horizontal sections were taken from three defined levels: Subocclusal (*so*); Central (*c*); Apical (*a*). (**c**) Magnification of the central horizontal section with indicated pulp horns (1–5). (**d**) Same image as shown in Figure 1b. The process of further sectioning and separation of pulp horns is displayed. (**e**) Final specimen before placement in embedding cassettes.

Collected dental sections were decalcified for 6 weeks in glass bottles (capacity 10 mL) using buffered EDTA (ethylenediaminetetraacetate), pH 8.0, at room temperature on a platform shaker (Polymax 1040, Heidolph Instruments, Schwabach, Germany). During this time, the EDTA solution was changed twice a week. After decalcification, the individual dental pulp positions were separated from each other using a scalpel. The trimmed samples measured approximately 15 mm × 10 mm × 5 mm and were placed in embedding cassettes (Simport™ Acetal Macrosette, Fischer Scientific GmbH, Schwerte, Germany) for further EDTA decalcification for an additional 2 weeks.

### 2.3. Sectioning and Staining

Following decalcification, the samples were rinsed using tap water and stored overnight in PBS (phosphate-buffered saline, Carl Roth GmbH, Karlsruhe, Germany).

Through the use of an automatic tissue infiltration machine (JungTP 1050, Leica Biosystems Nussloch GmbH, Nussloch, Germany), the samples were dehydrated through an ascending alcohol series and embedded in paraffin wax by an automatic embedding system (EG1150h, Leica Biosystems Nussloch GmbH, Nussloch, Germany). Subsequently, 7-micrometer-thin sections were cut on a slide microtome (RM2125RT, Leica Microsystems GmbH, Wetzlar, Germany) and dried overnight at 37 °C in an incubator (BE 200, Memmert GmbH & Co. KG, Schwabach, Germany). To ensure optimized adhesion, consecutive sections were mounted on SuperFrost Plus™ (Fischer Scientific GmbH, Schwerte, Germany) slides and stained with toluidine blue after a routine protocol.

### 2.4. Evaluation of the Histological Sections

Toluidine-blue-stained sections were examined by light microscopy (Leica DM2500, Leica Microsystems GmbH, Wetzlar, Germany). Using an automated picture-aligning tool (Leica LAS XY Live Image Builder, Leica Microsystems GmbH, Wetzlar, Germany), the entirety of each sampled pulp was photographed at 10× and/or 20× magnification. Subsequently, these images were morphometrically analyzed in terms of their expansion. Through manual outlining, the total area of the dental pulp was determined by a morphometric software (Leica LAS X Quantify, Stack Profile, ROI Area, Leica Microsystems GmbH, Wetzlar, Germany). Maximum length and maximum width were determined using a scale bar tool (Leica LAS X Quantify Draw Scalebar, Leica Microsystems GmbH, Wetzlar, Germany). Due to the fact that horizontal sections of the pulp did not form perfect geometrical shapes (circular or oval), the measurement lines were not always perpendicular to each other. Further, the width of the odontoblastic layer and the predentin were measured (Leica LAS X Quantify Draw Scalebar, Leica Microsystems GmbH, Wetzlar, Germany). Furthermore, the type of predentin mineralization (globular or linear) was recorded (Figure 3). Preliminary examinations on 50 pulp horns of different aged teeth on all defined hotizontal levels showed the radial symmetry of the pulp horns. Thus, single measurements of the width of the odontoblastic layer and the width of predentin were obtained.

## 3. Statistical Analysis

Descriptive statistics were performed with GraphPad Prism 6 (GraphPad Software, Inc., La Jolla, CA, USA). If components of inferential statistics were included in charts, they were initially calculated with SigmaStat 4.0 (Systat Softwares Inc., San José, CA, USA).

Data are presented as dot plots, whereby the arithmetical average is illustrated by a horizontal bar and the related standard deviation by vertical whiskers. For analyses of inferential statistics, SigmaStat 4.0 methods were applied. Pulp length, width, area, and the thickness of the odontoblastic layer and predentin were evaluated by a two-way analysis of variance (ANOVA) with repeated measurements. To repeat measurements, various section planes of one tooth were surveyed. One factor was represented by the age group, AG 1–3 (AG 1: 0–210 days, AG 2: 5–14 years, AG 3: 19–24 years) and the other by the section plane, *so* (subocclusal), *c* (central), or *a* (apical). Significant data in the variables of pulp length, width, and area were detected, and therefore, further analyses were performed by applying the post hoc test by Tukey. In cases lacking normal distribution, a logarithmic transformation and, once, a Johnson transformation, were applied. For all variables with exception of the pulp area, the comparison between incisors and cheek teeth was performed by a paired t test. If normal distribution was lacking, which was only the case for the pulp area, a Wilcoxon signed-rank test was implemented. In general, the significance level was determined at α ≤ 0.05. In diagrams, *p*-values are indicated as *p* ≤ 0.05 (*), *p* ≤ 0.01 (**), and *p* ≤ 0.001 (***).

## 4. Results

### 4.1. Length and Width

In the deciduous incisors, the horizontal length and width of the dental pulp in the foals (AG 1) were significantly longer and wider than in the adult horses, AG 2 (*p* < 0.001) and AG 3 (*p* < 0.001). Regarding the pulpal length along the assessed horizontal levels, there was a significant decrease from the subocclusal level (mean (m) = 5239 µm) to the central level (m = 3735 µm) and to the apical level (m = 3291 µm, *so* to *a*: *p* = 0.004, *c* to *a*: *p* = 0.028) in AG 1, but a constant pulpal length was recorded for AG 2 and AG 3. The pulpal width remained constant at all the sectioned levels (*so*, *c*, *a*) in all the AGs, showing decreasing absolute values from AG 1 (m = 2042 µm) to AG 2 (m = 812 µm) and to AG 3 (m = 476 µm).

In the cheek teeth, the dental pulps were also significantly longer and wider in AG 1 compared to AG 2 (length and width: *p* = 0.009) and AG 3 (length: *p* < 0.001, width: *p* = 0.002). In contrast to the incisors, the pulpal length and width in the cheek teeth of the AG 1 and AG 2 horses increased from the subocclusal level to the apical level. In AG 3, the pulpal length and width in the cheek teeth remained constant (Figure 4 and Figure 5, Table 2).

### 4.2. Total Area

In the incisors and the cheek teeth, the total measured area in the equine dental pulp decreased significantly from AG 1 to AG 3 (*p* < 0.001).

In the incisors of the foals (AG 1), the total pulpal area showed a biconical distribution, with the highest value (m = 7419 µm^2^) in the central aspect and the lowest values at the subocclusal (m = 6503 µm^2^) and the apical (m = 6520 µm^2^) levels. In AG 2, the total area of the incisors was significantly smaller compared to AG 1 (*p* = 0.001), showing increasing values from *so* (m = 571 µm^2^) to *c* (m = 838 µm^2^) and *a* (m = 994 µm^2^). The incisors in AG 3 showed the smallest total pulpal area compared to AG 1 (*p* < 0.001) and AG 2 (*p* = 0.004). Regarding the assessed horizontal levels, in AG 3, the total pulpal area remained constant.

Similar to the incisors of AG 2, the total pulp area in the cheek teeth increased in the apical direction in AG 1 (*so* to *a*: *p* = 0.021). In AG 3, the total pulpal area in the cheek teeth remained constant along the tooth axis (Figure 6, Table 3).

### 4.3. Odontoblastic Layer and Predentin

Concerning the widths of the odontoblastic layer and of the predentin, the values were in the same small range in the incisors and the cheek teeth. In all the age groups and at all the investigated levels, no statistically significant differences were recorded.

The odontoblastic layer measured between 0 and 60 µm, showing a high standard deviation (SD); the predentin width ranged from 0 to 20 µm (Figure 7). In two thirds of the horses, a linear and globular mineralization front in one pulp system was obtained, independently of age group and tooth type. One third of the horses showed globular or linear mineralization. Furthermore, no statistical differences were recorded regarding the predentin mineralization front in the different age groups and assessed horizontal levels.

## 5. Discussion

The results of the present study demonstrate the unique and clinically relevant characteristics of the equine hypsodont dentition. From an odontological perspective, the reported cellular characteristics of the pulp tissue support the hypothesis that the endodontium of the equine tooth remains in an immature and highly productive state over its entire lifespan. The results concerning the spatial dimensions of the pulp system provide useful and instructive information with regard to the further development of equine-specific endodontic treatments.

### 5.1. Morphology: Age-Related Changes in the Equine Dental Pulp

Our results show that, in the incisors as well as in the cheek teeth, the horizontal length, width, and total area of the dental pulp in the deciduous teeth of the foals were enlarged compared to the permanent teeth of the adult or even the older horses. These results are not surprising, as it is well understood that recently erupted teeth feature wide pulp cavities, which are not yet diminished by the deposition of secondary dentin [1,4,9,18,22,30].

In the incisors of the foals, the horizontal expansion of the dental pulp tapered from the occlusal level to the apical level [8]. Meanwhile, the horizontal expansion of the pulp in the incisors of the adult and senior horses (AG 2 and AG 3) was nearly constant, forming a straw-like shape [22]. The same phenomenon occurred within the dental pulp of the cheek teeth of the AG 3 horses. In contrast to the latter finding, the horizontal pulpal expansion in the cheek teeth of the AG 1 and AG 2 horses was enlarged in the apical direction, forming a cone-like shape. This can be explained by the intercommunicating pulp horns in the deciduous and young permanent teeth, creating a common pulp chamber in the apical region in the cheek teeth [8,31]. Because of the continued deposition of secondary dentin, the common pulp chamber began to split up into separate pulp compartments, composed of pulp horns and root canals, in the aged teeth. These age-related changes in the pulp chamber are not unusual and have been described by several authors [18,25,32].

Our data do not display the three-dimensional shape of the pulp system as only selected two-dimensional, horizontal sections were investigated. However, our data add information on the exact dimensions of the pulp compartment and, therefore, complete the three-dimensional data presented in previous studies [25,27,32]. In contrast to equine hypsodont teeth, a common pulp chamber is placed in the crown region in the brachydont teeth of humans, which gives rise to pulp canals in the apical direction [14,16]. Using cone-beam computed tomography (CBCT), the dimensions of the pulp chamber have been determined in human teeth [33,34]. The pulp chambers in young human teeth feature cross-sectional diameters of up to 10 mm^2^. With age, these cross-sectional diameters decrease to 3 mm^2^ [33]. In our study, the highest mean area values were measured within AG 1. In the incisors, a mean area of up to 7.4 mm^2^ was detected; in the cheek teeth, the mean area was up to 6.8 mm^2^. Thus, the dimension of the pulpal compartments in horse teeth is similar or, surprisingly, even smaller than that in brachydont teeth, although the occlusoapical lengths of equine teeth are much greater.

The documented dimensions of equine pulp compartments, the overall length of equine teeth, and the complex architecture of the equine pulp system [25,35] should be considered in equine-specific endodontic treatments. Detailed knowledge of the spatial dimensions and architecture of the pulp compartments is of central importance for modern endodontic treatment approaches in human dental medicine. Therefore, it has been recommended to display the three-dimensional anatomy of the pulp systemthrough the use of cone-beam computed tomography prior to endodontic treatment [34,36,37].

### 5.2. Cellular Arrangement: Constant Odonotblastic Layer and Predentin

Although the dimensions of equine dental pulp change with age and differ between the incisors and cheek teeth, the cellular structure responsible for the continued production of secondary dentin, the odontoblastic layer, generally remain constant. In contrast to our findings, the aged teeth of humans show changes in the size and appearance of their odontoblastic layers. In brachydont teeth, odontoblasts transform from actively producing columnar cells in younger age groups to flattened, resting cells in older age groups, which have lower dentin production rates [14,38,39]. Moreover, a significant decrease in odontoblastic cell density, of 50–75%, has been reported for brachydont teeth from the age of about 50 years [23,40,41].

In horses, the prolonged eruption of hypsodont teeth requires the life-long preservation and activity of the odontoblasts to prevent occlusal pulp exposure, as these teeth experience significant dental wear. Assuming that equine teeth lose 3–4 mm of dentin at their occlusal surface per year, the compensatory production of a similar amount of dentin at the occlusal aspect of the pulp horns is required to prevent pulpal exposure. Thus, a production rate of subocclusal dentin of approximately 8–10 µm per day is necessary, even in older horses. This calculated rate corresponds to the dentin production demonstrated for non-erupted, i.e., developing, brachydont teeth [42,43].

Considering the width of the predentin layer, no age-related changes were detected in our observations. The mean widths of the predentin layer in the incisors and cheek teeth ranged from 7–14 µm, which corresponds to the estimated production rate that is required for equine teeth to prevent pulpal exposure. Two thirds of the examined horses in this study showed both a linear and a globular mineralization front in one pulp horn at the same time. This histomorphological feature indicates a wave-like, long-term odontoblastic activity since, as is well known, globular mineralization occurs in fast mineralization processes, while linear mineralization is correlated with slower mineralization processes [14,44]. In brachydont teeth, a surprisingly age-related increase occurs in the width of the predentin layer from a mean value of 14–20 µm in teeth with incomplete root formation to a mean value of 57 µm in humans over 30 years. However, this surprising finding is not explained by accelerated predentin production but, rather, by diminished mineralization rates [45,46].

## 6. Conclusions

According to the obtained findings, it is assumed that the cellular components of equine dental pulp remain in a juvenile and active state, presumably triggered by the constant biomechanical forces generated by occlusal wear. However, due to the limited number of investigated specimens, these findings and assumptions should be regarded as preliminary, albeit clinically significant. The proposed consistency of the odontoblastic activity does not only prevent pulp exposure at the occlusal surface under physiological conditions, but also allows it to react efficiently in terms of tertiary dentin production under pathological conditions.

Endodontic treatments might utilize the high productivity of equine pulpal cells to stimulate the production of intra-pulpal dentinal bridges to demarcate diseased pulpal areas for vital pulp regions. However, access to and manipulation within equine dental pulp might be complicated due to the delicate dimensions of the equine pulp system elucidated in this study.

## 7. Limitations

Before the actual histomorphometric evaluation started, a procedure had to be established to prepare the different tooth sections because of the need for a proper decalcification process. The less-decalcified specimens increased the risk of disruption artefacts in the histological sections, as well as the use of an automated ultrasound decalcifier. With the method of image building, we were able to create a histological, high-resolution overview image of the equine dental pulp for an optimal evaluation of the datasets. Thus, the number of subjects had to be kept low, as both methods require a large amount of time. Further, it must be borne in mind that angular deviations in the horizontal sections could have affected the measured values.

## Figures and Tables

**Figure 1 vetsci-09-00261-f001:**
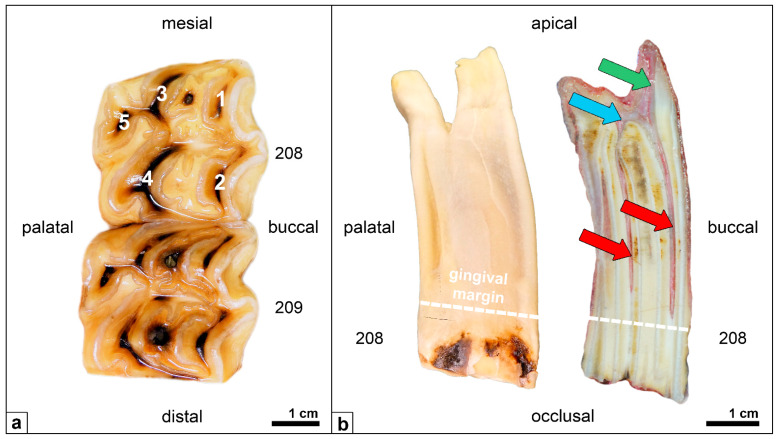
(**a**) Upper-cheek teeth of an adult horse (208, 209), occlusal view; pulp positions (1–5) appear as dark brown spots. (**b**) Cheek tooth (208), mesial view (**left**) and longitudinal section (**right**) Red arrows: pulp horns. Blue arrow: common pulp chamber. Green arrow: root canal.

**Figure 3 vetsci-09-00261-f003:**
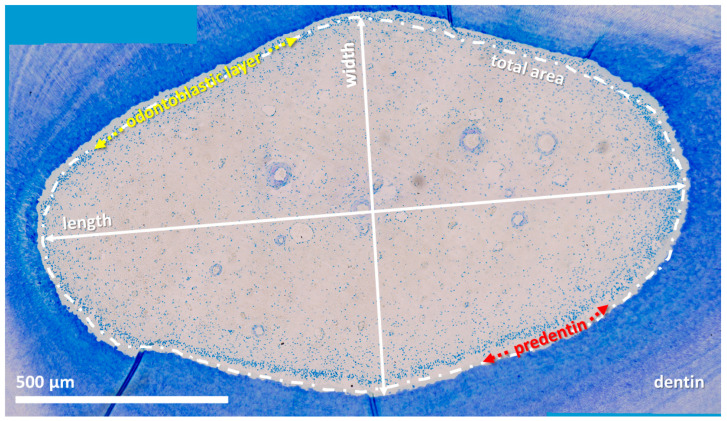
Horizontal histological section (subocclusal level) of an incisor (203) from a 22-year-old horse, toluidine blue stain. Approximately 100 single high-resolution images (magnification: 20×) were aligned using LAS X, XY Live Image Builder.

**Figure 4 vetsci-09-00261-f004:**
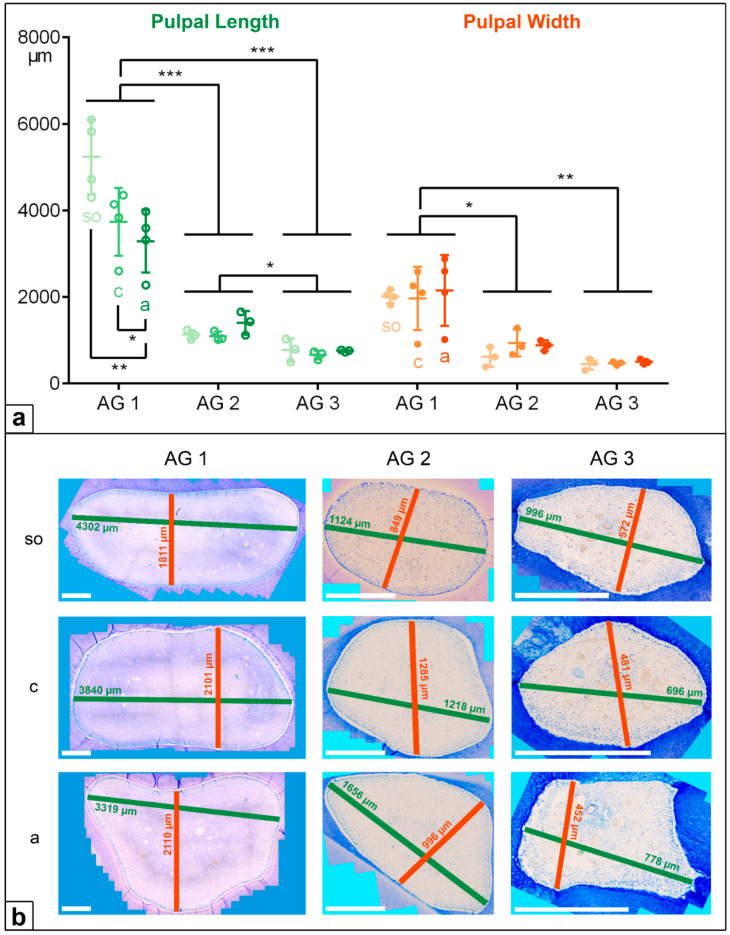
(**a**) Dot-plot diagram showing the pulpal length (green colours) and width (orange colours) of incisors of different age groups (AG). Within each age group, the following horizontal levels are shown: *so* (subocclusal), *c* (central), and *a* (apical). Mean: central horizontal bar. Standard deviation: vertical whiskers. *p*-values are indicated as *p* ≤ 0.05 (*), *p* ≤ 0.01 (**), and *p* ≤ 0.001 (***). (**b**) Figures presenting the horizontal expansion of the dental pulp in incisors regarding different age groups (AG) and different horizontal levels (*so*, *c*, *a*). Scale bar: 500 µm.

**Figure 5 vetsci-09-00261-f005:**
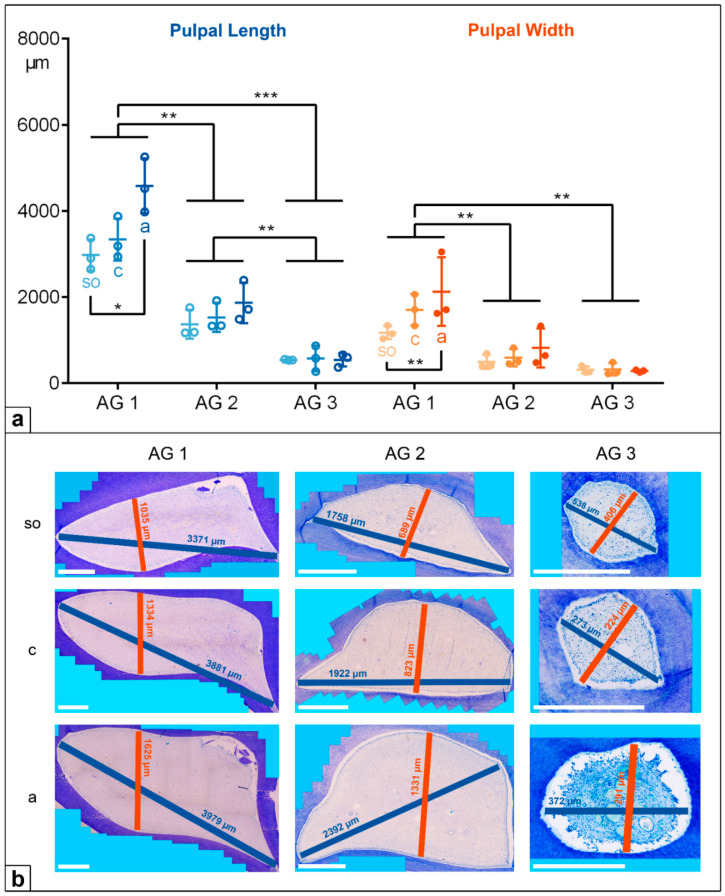
(**a**) Dot-plot diagram showing the pulpal length (blue colours) and width (orange colours) of cheek teeth of different age groups (AG). Within each age group, the following horizontal levels are shown: *so* (subocclusal), *c* (central), and *a* (apical). Mean: central horizontal bar. Standard deviation: vertical whiskers. *p*-values are indicated as *p* ≤ 0.05 (*), *p* ≤ 0.01 (**), and *p* ≤ 0.001 (***). (**b**) Figures presenting the horizontal expansion of the dental pulp in incisors regarding different age groups (AG) and different horizontal levels (*so*, *c*, *a*). Scale bar: 500 µm.

**Figure 6 vetsci-09-00261-f006:**
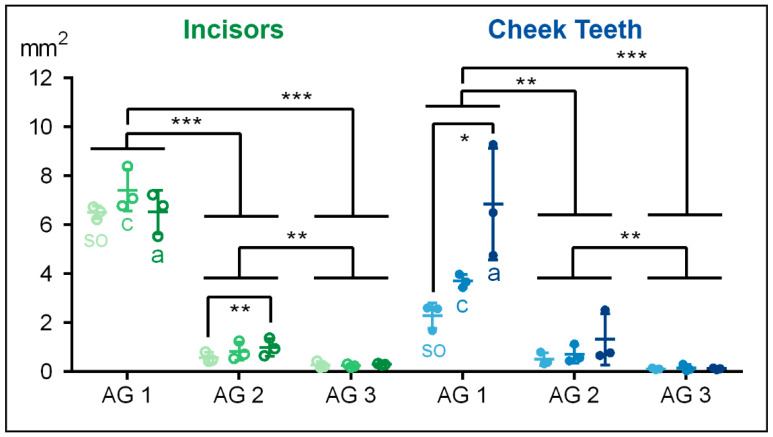
Dot-plot diagram showing the total areas of the incisors (green colours) and cheek teeth (blue colours) of different age groups (AG). Within each age group, the following horizontal levels are shown: *so* (subocclusal), *c* (central,) and *a* (apical). Mean: central horizontal bar. Standard deviation: vertical whiskers. *p*-values are indicated as *p* ≤ 0.05 (*), *p* ≤ 0.01 (**), and *p* ≤ 0.001 (***).

**Figure 7 vetsci-09-00261-f007:**
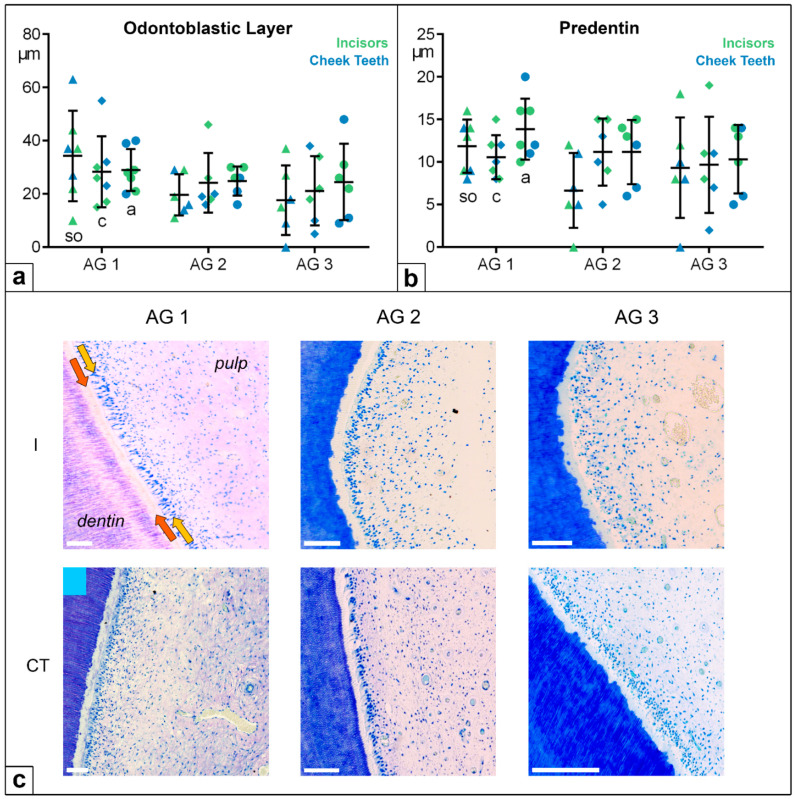
(**a**) Dot-plot diagram showing the widths of the odontoblastic layers of incisors (green colours) and cheek teeth (blue colours) of different age groups (AG). Within each age group, the following horizontal levels are shown: *so* (subocclusal) shown as triangle, *c* (central) shown as diamonds, and *a* (apical) shown as circles. Mean: central horizontal bar. Standard deviation: vertical whiskers. (**b**) Dot-plot diagram showing the width of the predentin of incisors (green colours) and cheek teeth (blue colours) of different age groups (AG). Within each age group, the following horizontal levels are shown: *so* (subocclusal), *c* (central), and *a* (apical). Mean: central horizontal bar. Standard deviation: vertical whiskers. (**c**) Figures presenting the dentin–pulp complex in incisors (I) and cheek teeth (CT) regarding different age groups (AG) at the central level (*c*). Orange arrow: predentin. Yellow arrow: odontoblastic layer. Scale bar: 50 µm.

**Table 1 vetsci-09-00261-t001:** Data of horses examined.

No.	Sex	Breed	Age (Days/Years)	Age Group (AG)	Tooth Sample	Cause of Euthanasia
1	mare	Warmblood	2 d pre-part.	1	801	abortion
2	mare	Black Forest draft horse	2 d	1	608, 701	colic
3	stallion	Warmblood	5 d	1	608	septicemia
4	mare	Warmblood	40 d	1	608, 801	colic
5	mare	Shetland pony	210 d	1	608, 801	colic
6	mare	Pony	5 y	2	208, 301	dystocia
7	gelding	Warmblood	12 y	2	208, 401	colic
8	mare	Warmblood	14 y	2	208, 401	atypical myopathy
9	mare	Icelandic horse	19 y	3	208, 301	colic, septicemia
10	mare	Warmblood	19 y	3	401	colic
11	mare	Warmblood	21 y	3	108, 401	colic
12	gelding	Pony	24 y	3	108	laminitis

**Table 2 vetsci-09-00261-t002:** Additional statistical significances of lengths and widths of incisors and cheek teeth.

	Incisors	Cheek Teeth
Length	AG 1 vs. AG 2: *p* < 0.001	AG 1 vs. AG 2: *p* = 0.009
	AG 1 vs. AG 3: *p* < 0.001	AG 1 vs. AG 3: *p* < 0.001
	AG 2 vs. AG 3: *p* = 0.012	AG 2 vs. AG 3: *p* = 0.003
Width	AG 1 vs. AG 2: *p* = 0.011	AG 1 vs. AG 2: *p* = 0.009
	AG 1 vs. AG 3: *p* < 0.001	AG 1 vs. AG 3: *p* = 0.002
	AG 2 vs. AG 3: no	AG 2 vs. AG 3: no

**Table 3 vetsci-09-00261-t003:** Additional statistical significances of total area in incisors and cheek teeth.

	Incisors	Cheek Teeth
Total Area	AG 1 vs. AG 2: *p* = 0.001	AG 1 vs. AG 2: *p* = 0.004
	AG 1 vs. AG 3: *p* < 0.001	AG 1 vs. AG 3: *p* < 0.001
	AG 2 vs. AG 3: *p* = 0.004	AG 2 vs. AG 3: *p* = 0.003

## Data Availability

The data presented in this study are available on request from the corresponding author.

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
