# Peer review of "The Equine Dental Pulp: Histomorphometric Analysis of the Equine Dental Pulp in Incisors and Cheek Teeth"

_vetsci, 2022, doi:10.3390/vetsci9060261_

Round 1

Reviewer 1 Report

The authors provided a concise and well written manuscript that characterizes the length width and area of dental pulp in one incisor and one cheek tooth in horses grouped into three age categories. The authors also included data measuring the width of the odontalblast and predental layers in these teeth.

The major limitation in the study is the sample size, the authors attempt to justify that limitation by saying time was a limiting factor (line 314-315). That may be the case, but is not a justifiable reason to not provide enough data. Each animal used in this study had more teeth and more animals could have been used. Fixed rather than fresh tissue were used further preventing time as a limitation in study design. If the authors intend not to add additional data the authors should indicate this as a preliminary or pilot study. Alternatively they are encouraged to take the time to generate a more comprehensive dataset.

To that same point only one incisor and one cheek tooth was sampled from each animal. These were the same tooth (x01, x08) presumably because the authors acknowledge that their may be variation between teeth. Why it that the authors did not look at multiple incisiors and cheek teeth is not clear. The authors are encouraged to offer evidence of the representative nature of the selected incisor and cheek tooth (to all other teeth of that type) or they should edit the manuscript to remove extrapolations beyond the characteristic of x01, and x08 specifically.

The authors must revise the age grouping definitions in this study as they include animals that are not represented in the dataset. AG1 could be considered 0-210days, AG 2 as 5 – 14 years, and AG3 as 19-24 years. Currently the authors only have 3-4 samples in each of these cohorts, and no samples from 210 days- 5 years, and from 14-19 years.

Additional detail on how the maximum length and width were determined should be included in this manuscript. Width was not placed perpendicular to length. The representative images do not show a consistent indication of how these were identified.

Details on how area was measured and/or derived should be included.

Where and how the measurements of odonotoblast and predental layers were made must be included. Were multiple measurements made and averaged per tooth or a single measurement obtained? From what side of the tooth?

Graphs should be labeled to indicate length and width (figure 3a, 4a); tooth x01 and x08 (figure 5) and odonotoblast and predental layers (Figure 6 a,b).

The conclusions should be rewritten as they draw conclusions that are not supported by the content of this manuscript (including cellular activity). The authors are should confine their conclusions to those that can be made based on the descriptive (non-functional data) reported in this study.

References (like Lundstrom 2016) should be revised so that all references are formatted consistently and in accordance with journal standards.

Author Response

Dear Reviewer,

With kind regards, Jessica Roßgardt 

Reviewer 2 Report

Dear Authors, the manuscript with the title "The equine dental pulp: histomorphometric analysis of the equine dental pulp in incisors and cheek teeth" is an excellent study and the results are very important for continuing education of practitioners of equine dental surgery.

Author Response

(The authors gave the same response as above.)

Reviewer 3 Report

Thank you for the opportunity to review this manuscript describing histomorphometric analysis of equine dental pulp in incisors and cheek teeth. The methods are interesting and the figures complement the manuscript well. However, overall, the discussion of findings appears disjointed and the potential clinical impact as written is not well emphasized and could be highlighted further. See specific comments below.

27-40 – recommend combining first two paragraphs

35-36 – consider rewording for clarity to: Equine incisors and cheek teeth contain up to two or seven pulp positions, respectively, which appear in a highly constant arrangement.

39 – typically spell out numbers less than 10 – ‘one’

58-62 – combine into one paragraph at end of introduction

Introduction in general would be supported by a diagram/figure illustrating the anatomy described.

73 – how was clinical health determined? Was this determined antemortem or postmortem?

Perhaps I am missing it, but how many horses were in each age group and were ages evenly distributed? Were horses enrolled convenience sampled as they were euthanized for other reasons?

151 – statistical analysis indicates significance was set at p<0.05 but figures seem to indicate differential p-values indicated by *, **, *** - either change figures to include only * indicating p<0.05, or define what p-values are indicated by *,**,*** etc.

Figure 3,4,5,6 – recommend including legends of ‘pulpal length’ and ‘pulpal width’ to the itself to improve readability

Discussion seems to suggest in several places that the authors’ findings have been previously published (231-232; 243-244) – recommend emphasizing what their findings add to the body of literature / previous work that has not already been reported.

246-260 – appears somewhat disjointed as broken into multiple single line paragraphs – can these be combined into one paragraph in a cohesive manner? Overall, the discussion does not read smoothly as written – recommend revision into paragraphs with single overriding discussion point summarized in the first sentence of each paragraph.

Conclusions – recommend further expansion on the clinical applicability of these findings – overall, the main goal of this work, beyond description of anatomy, is not clearly stated.

Author Response

(The authors gave the same response as above.)

Round 2

Reviewer 1 Report

Thank you for addressing the comments to some extent. I would prefer the pilot/preliminary nature of the findings be included in the title and throughout. I would also like additional detail on how limits to the sub-layers were measured, but I find the current version adequate for publication.

Author Response

Dear Reviewer,

see the author's answers in the attachment.

With kind regards,

Jessica Roßgardt

Reviewer 3 Report

The authors appear to have thoughtfully replied to each of the reviewer's comments and the manuscript is improved in its current format. One minor spelling error in the discussion, 5.1 'Morphology' not 'Morpholgy.' Thank you to the authors for their responses and efforts.

Author Response

Dear Reviewer,

please see the author's answers in the attachment.

With kind regards,

Jessica Roßgardt
